# Unmasking the Metabolite Signature of Bladder Cancer: A Systematic Review

**DOI:** 10.3390/ijms25063347

**Published:** 2024-03-15

**Authors:** Francisca Pereira, M. Rosário Domingues, Rui Vitorino, Inês M. S. Guerra, Lúcio Lara Santos, José Alexandre Ferreira, Rita Ferreira

**Affiliations:** 1LAQV-REQUIMTE, Mass Spectrometry Centre, Department of Chemistry, University of Aveiro, 3810-193 Aveiro, Portugal; afranciscagp@ua.pt (F.P.); ines.guerra@ua.pt (I.M.S.G.); ritaferreira@ua.pt (R.F.); 2Experimental Pathology and Therapeutics Group, IPO Porto Research Center (CI-IPOP), RISE@CI-IPOP (Health Research Network), Comprehensive Cancer Center, Portuguese Oncology Institute (IPO Porto), 4200-072 Porto, Portugal; llarasantos@gmail.com (L.L.S.); jose.a.ferreira@ipoporto.min-saude.pt (J.A.F.); 3CESAM, Mass Spectrometry Centre, Department of Chemistry, University of Aveiro, 3810-193 Aveiro, Portugal; mrd@ua.pt; 4iBiMED, Department of Medical Sciences, University of Aveiro, 3810-193 Aveiro, Portugal

**Keywords:** bladder cancer, metabolomics, biomarkers, mass spectrometry, nuclear magnetic resonance, urine, blood-derived samples, tumor tissue

## Abstract

Bladder cancer (BCa) research relying on Omics approaches has increased over the last few decades, improving the understanding of BCa pathology and contributing to a better molecular classification of BCa subtypes. To gain further insight into the molecular profile underlying the development of BCa, a systematic literature search was performed in PubMed until November 2023, following the PRISMA guidelines. This search enabled the identification of 25 experimental studies using mass spectrometry or nuclear magnetic resonance-based approaches to characterize the metabolite signature associated with BCa. A total of 1562 metabolites were identified to be altered by BCa in different types of samples. Urine samples displayed a higher likelihood of containing metabolites that are also present in bladder tumor tissue and cell line cultures. The data from these comparisons suggest that increased concentrations of L-isoleucine, L-carnitine, oleamide, palmitamide, arachidonic acid and glycoursodeoxycholic acid and decreased content of deoxycytidine, 5-aminolevulinic acid and pantothenic acid should be considered components of a BCa metabolome signature. Overall, molecular profiling of biological samples by metabolomics is a promising approach to identifying potential biomarkers for early diagnosis of different BCa subtypes. However, future studies are needed to understand its biological significance in the context of BCa and to validate its clinical application.

## 1. Introduction

Cancer of the urinary bladder, or bladder cancer (BCa), is the most common malignancy of the urinary tract, being the 10th most incident malignant disease worldwide. In 2020, there were more than 573,000 BCa cases and over 212,000 deaths globally [1]. Moreover, BCa comprises several risk factors, such as gender, age, chemical and environmental exposures and genetic and molecular abnormalities, among others [2,3,4,5,6]. The clinical presentation of BCa depends on the stage and grade of the malignancy. Hematuria, frequent urination, nocturia and dysuria are non-specific signs of lower urinary tract diseases [7]. Given that these symptoms and signs are not exclusive of a BCa case, its diagnosis relies on a series of tests, which mainly include urine analysis, cytology and cystoscopy [8].

The BCa classification takes into account histological and pathophysiological features that are essential for adequate staging and classification of the malignant disease. In terms of histomorphology, urothelial cell carcinoma (UCC) is the most common subtype (~75%), followed by squamous cell carcinoma (SCC) and adenocarcinoma [9]. In terms of pathophysiology, the staging of BCa follows the tumor–node–metastasis (TNM) system, with non-muscle-invasive bladder carcinoma (NMIBC) confined to the urothelium or lamina propria (stage T1) and muscle-invasive bladder carcinoma (MIBC) invading the muscularis propria (stage T2), perivesical tissue (stage T3) or adjacent organs and pelvic structures (stage T4). The staging is based on the classification of the World Health Organization (WHO), which distinguishes between papillary urothelial neoplasms with low malignant potential and low-grade and high-grade papillary carcinomas [10,11]. Accurate staging and grading of tumor characteristics is crucial for the follow-up of every BCa patient.

At the molecular level, BCa comprises several subtypes, including luminal–papillary, luminal–infiltrated, basal–squamous and neuronal, among others. Each subtype is characterized by unique molecular and pathological features [12,13,14]. However, due to the heterogeneity of this malignancy, it is difficult to reach a consensus on the molecular classification of BCa. In an effort to gain a better insight into tumor biology, BCa research has therefore increasingly resorted to Omics approaches in recent decades. As “deregulation of cellular metabolism” is considered one of the hallmarks of cancer [15], metabolomics research in BCa has proven particularly valuable for studying the pathophysiology of BCa. This research led to the identification of metabolite signatures that reflect the molecular processes underlying the different BCa subtypes [16]. Moreover, metabolomic profiling has allowed the identification of potential biomarkers, not only for a better molecular classification but also to improve the understanding of the alterations underlying BCa development [17,18,19]. In the pursuit of promising strategies, metabolomic-based approaches applied to the analysis of non-invasively, or minimally invasively, collected body fluids have been demonstrated to be specific and cost-effective tools for the discovery of novel and specific biomarkers for early detection and diagnosis of BCa and high-risk patients, along with improved survival and interpretation of the disease status [20,21,22,23]. Moreover, efforts have been made to integrate data from metabolomics studies to propose biomarkers for BCa management [24,25,26,27,28]. In the present review, we aimed to delve deeper into the understanding of the molecular changes occurring during BCa development through an updated systematic literature search of experimental studies focusing on BCa metabolite profiling, followed by an integrative analysis using bioinformatic tools. Data from this analysis undergoes critical discussion of the reported data. A list of altered metabolites associated with the detection, development and prognosis of BCa that can be considered as potential biomarkers is presented.

## 2. Material and Methods

### 2.1. Search Strategy and Selection Criteria

A systematic literature search in the PubMed database was performed and conducted according to PRISMA guidelines. The last search was run on the 9 November 2023, and it was restricted to the English language. The search included experimental studies based on the following keywords: “bladder cancer” AND “lipidomics” OR “metabolomics” AND “blood”, “urine”, “cell line” OR “tissue”. As a result, the search retrieved a total of 261 published studies, which were considered if they were related to the domains of BCa and metabolomics-based science data. Following title and abstract analysis, 87 papers were selected, and after full-text and results analysis, 25 experimental studies were pooled, all published between 2012 and 2023 (Figure 1). Experimental studies were included if they reported altered metabolites involved in BCa. The study selection was performed by two individuals independently, and disagreements were resolved by consensus between the authors or by a third author [Appendix A]. From these, 3 performed metabolomic profiling of BCa cells, 4 of BCa human and mice-derived tissue samples and 18 analyzed body fluids, in which 10 included exclusively analysis of urine samples, 7 included only the analysis of blood samples and 1 included both urine and blood samples. 

### 2.2. Analysis

The analysis of the selected papers resulted in a list of 1562 altered metabolites in BCa, presented in Appendix A. The metabolite names and other metadata from the original paper were retained whenever possible. In cases where this information was not provided, a search was conducted in PubChem and HMDB. Furthermore, the trends in metabolite variation outlined in this table correspond with those reported in the original papers. Metabolites modulated by BCa were identified by comparing data across various sample types using the Venny tool (https://jvenn.toulouse.inrae.fr/app/example.html), considering the names of the metabolites. For the metabolites consistently modulated by BCa in urine, bladder tissue and cell lines, a MetScape analysis (v3.1.3.; http://metscape.ncibi.org) was conducted. Out of the nine metabolites identified as modulated by BCa, only six were included in this analysis due to the availability of their KEGG IDs. The size of the nodes reflects variations in the metabolite levels; they appear enlarged when at higher levels and smaller when at lower levels in samples from subjects with BCa.

## 3. Results and Discussion

### 3.1. Overview of the Methodological Approaches Used in Bladder Cancer Metabolomics Research

In the field of oncology, metabolomics-based research uses two main analytical techniques for metabolite characterization, namely nuclear magnetic resonance (NMR) and mass spectrometry (MS). Each of these technological platforms has its own characteristics, advantages and disadvantages. The general workflow used for metabolomics-based research in the context of BCa is overviewed in Figure 2.

The selection of the appropriate methodology for metabolite extraction is dependent on the analytical system used, as well as the sample matrix, and it should be adapted according to the required metabolite range [29]. The majority of experimental studies that performed an LC-MS-based analysis included the use of cold organic solvents, such as acetonitrile [18,22,30,31] or methanol [17,20,23,32], for deproteinization prior to separation by centrifugation. In addition, metabolite extraction may also include the use of a mixture of solvents, such as acetone and methanol or acetonitrile and methanol [22,33,34]. Meanwhile, two studies that performed a GC-MS-based analysis used polydimethylsiloxane/divinylbenzene for the solid-phase microextraction of metabolites [35,36]. In the case of profiling, particularly for the lipid compounds, the workflow starts with total lipid extraction from the biological samples, using usually organic solvents. Several extraction methods can be performed, such as the Bligh and Dyer method and the Folch method, which use chloroform/methanol in a ratio of 1:2 and 2:1 by volume, respectively [37]. From the selected experimental papers [Appendix A], the Bligh and Dyer method was applied in two distinct studies [38,39], whereas the Folch method was used in the study of Ho and colleagues [40]. After metabolite extraction, the appropriate analytical system for metabolite detection is used. This can be performed using either NMR- or MS-based approaches, but the latter are more commonly used for BCa metabolomics due to their higher sensitivity and specificity, as well as their ability to identify a wide range of metabolites [41,42]. However, NMR can provide a non-specific and non-destructive quality and quantitative characterization of metabolites, with minimal sample preparation required [42,43]. Within the 25 selected experimental studies for this work (Figure 1), only three relied on NMR-based approaches [44,45,46], while the majority used MS-based ones, often in combination with separation techniques such as LC and GC. GC-MS was used in 4 out of 25 selected papers, identifying 124 potentially altered volatile metabolites in association with BCa pathophysiology [20,35,36,47]. Overall, while both NMR and MS-based approaches have their advantages and disadvantages, MS-based methods are more commonly used in BCa metabolomics due to their higher sensitivity and specificity.

For the identification of metabolites, databases for Omics-based data have been developed, e.g., TCGA, the Human Metabolome Database, PubChem and LipidMaps [41]. Once a list of the possible compounds in a biological sample is available, a statistical analysis is performed to obtain important biological information about the molecular profile of the sample [48]. However, to fully understand the altered metabolite signature in BCa, the data should be integrated and contextualized with clinical information from BCa patients and combined with data from different metabolomics platforms and bioinformatics tools. This approach could enable a better understanding of the biological significance of the altered metabolite signatures in BCa patients and ultimately lead to a better clinical outcome.

### 3.2. Data from Metabolomics Profiling in Bladder Cancer

BCa research using a metabolomics-based approach has been pursued through the use of three main categories of samples, namely human fluids, in vitro cell lines and ex vivo tissues [48,49]. Each type of sample has its own unique features, advantages and disadvantages.

For human fluids, both urine and blood samples are used in BCa research. Urine samples are used more frequently because this body fluid can reflect the molecular changes that occur during cancer development very well due to its direct contact with the developing tumor [50]. In this case, the disadvantage is that urine can be more affected by lifestyle and environmental factors as well as microbial contamination [51]. Furthermore, in contrast to blood samples, urine is easy, non-invasive and painless to collect, available in larger quantities and requires less sample preparation [48]. Nevertheless, blood is also a relevant sample as its composition reflects the molecular processes occurring throughout the organism, and it is less influenced by external factors [50]. In the case of cell lines, they have been widely used in cancer research as a result of their capability to provide an unlimited self-replicating source of biological material. Additionally, they can provide an alternative to animal testing, and their characteristics are available on online databases, providing a better insight into the genetic profile and the alterations associated with each BCa molecular subtype [49,52]. On the downside, the extrapolation of in vitro to in vivo model systems is still a challenging concern in science research [53]. Another drawback related to the use of in vitro cell lines is the fact that, when the culture lasts for a long period of time, they may be prone to the acquisition of epigenetic and genetic alterations [49]. Lastly, ex vivo bladder tissue samples imply a more direct study approach, providing information closely related to the tumor and its microenvironment; however, it is also a more invasive approach requiring certain expertise and special equipment, as well as careful, attentive and difficult sample preparation due to the sample heterogeneity [48].

In the next sections, the metabolite profile of each type of sample is critically analyzed in the setting of BCa.

#### 3.2.1. Data from Studies Performed with Urine-Derived Samples

Among the selected experimental studies, there are 11 that relied on the characterization of urine samples for the study of the molecular alterations that occur in BCa. These resulted in a list of 364 metabolites altered in urine-derived samples [Appendix A]. Most of these studies established a comparison between BCa patients and healthy controls and used liquid chromatography-MS (LC-MS) for the characterization of urine metabolites. In 2014, Jin et al. obtained the urine metabolomic profile of 138 BCa patients and found 12 metabolites that were altered when compared to control groups [54]. In the same year, a different study comprising 66 BCa patients identified a biochemical signature of 26 metabolites as potential biomarkers for this malignancy [23]. Interestingly, from the 38 metabolites found to be altered in the BCa cases of both studies, succinate was a common metabolite; however, it was found to be both increased [54] and decreased [23]. A recent study has also demonstrated decreased levels of succinate in urine-derived samples from BCa patients [46]. Succinate is an intermediate metabolite in the tricarboxylic acid (TCA) cycle that was demonstrated to play a role in inflammation, oxidative stress, ischemia, hypoxia and immune signaling [55,56,57]. Apart from these, succinate has been reported to promote tumorigenic signaling and metastasis, being defined as an oncometabolite [58,59]. Therefore, the increased content of succinate in urine samples from BCa patients may be an indicator of cancer cell migration and invasion. Afterward, in 2015, a study by Shen and colleagues found three BCa-upregulated—i.e., nicotinuric acid, trehalose and AspAspGlyTrp—and three BCa-downregulated—inosinic acid, ureidosuccinic acid and GlyCyzAlaLys—metabolites with high potential for BCa discrimination [60]. In this study, the alterations may be associated with an early diagnosis, given that the patients were mainly in stage T1. Nicotinuric acid is an acylglycine, a minor metabolite of fatty acid oxidation (FAO), and its increased content has been associated with the diagnosis of metabolic disorders [61]; however, the study performed by Shen and colleagues was the first to report it as a metabolite with increased levels in BCa. On the other hand, trehalose is a disaccharide that has been highly associated with antioxidant activity and is being investigated as a potential therapeutic agent to control cellular homeostasis in cancer cells [62]. Therefore, the increased content of this metabolite may be a consequence of increased oxidative stress and inflammation that occur during BCa development. Moreover, this disaccharide has recently been suggested as a potential prognostic serum biomarker for NMIBC [63]. Regarding the downregulated metabolites observed in this study, inosinic acid and ureidosuccinic acid are related to purine and pyrimidine metabolism, respectively. With that in mind, it would be expected that these metabolites would be increased due to the disturbed nucleotide metabolism of cancer cells to achieve an increased proliferation rate [64]. Therefore, the modulation of both inosinic and ureidosuccinic acid during BCa development requires further investigation. At last, urine samples of BCa patients have also been demonstrated to alter two tetrapeptides, namely AspAspGlyTrp and GlyCysAlaLys. Yet, the involvement of these compounds in cancer development, and especially BCa pathogenesis and progression, is poorly studied and requires further investigation.

As potential biomarkers for BCa diagnosis, a different study discovered 19 metabolites in significantly different amounts, in which a subset of 11 metabolites were filtered with 95% sensitivity and 100% specificity [17]. This study provided the detailed results of a metabolic pathway enrichment analysis of the 19 differently abundant metabolites. This allowed the identification of 33 distinct metabolic pathways, including fatty acid (FA) metabolism, propanoate metabolism, phenylacetate metabolism, proline and arginine metabolism, bile acid biosynthesis and others [17]. Among the altered metabolites, it is possible to point out adenosine monophosphate (AMP), chenodeoxycholic acid (CDCA) and glycochenodeoxycholic acid (GCDCA) as increased metabolites found in urine samples derived from BCa patients. AMP is a purine nucleotide that may lead to the activation of the AMP-activated protein kinase (AMPK) in conditions of low energy status, activating catabolic pathways to increase the generation of ATP [65]. That is, AMPK activation advantages tumor cell growth and proliferation. In addition, the measurement of this nucleotide has exhibited a correlation between AMP concentration and tumor progression, revealing high AMP levels in BCa cases that became invasive and/or metastatic [66]. The metabolites CDCA and GCDCA are a part of the bile acids group synthesized from cholesterol in the liver. Bile acids have double-regulatory roles in carcinogenesis, demonstrating both anti- and pro-carcinogenic activity depending on the neoplasia [67,68]. These compounds are active G-protein-coupled receptors that induce signaling pathways involved in inflammation, proliferation, migration and cancer development [68]. Notwithstanding, the outcome of these effects is dependent on local bile acid concentration, along with differential expression of its receptors between cancers. Hence, bile acid concentration in the context of BCa proliferation and migration requires further investigation. Concerning the decreased metabolites in the study of Wang and coworkers, a decrease in the content of picolinic acid was observed [17]. Picolinic acid is a product of tryptophan catabolism, which is produced in response to an inflammatory condition. It possesses a particular role as a second signal in the activation of macrophages [69,70]. Therefore, the decreased levels of picolinic acid found in this study suggest that during BCa pathology, the inflammatory response is likely decreased. This outcome is questionable given that an increased inflammatory response is a well-known hallmark of cancer. In addition, the levels of tridecanoic acid and myristic acid in urine samples of BCa patients were decreased in comparison to healthy controls. FA metabolism has been related to immunotherapy in BCa, in the sense that cases that exhibited low FA synthase expression demonstrated an increase in tumor sensitivity to drug therapy with immune checkpoint inhibitors [71]. Therefore, the downregulation of FA metabolism may possess a prognostic risk score value for BCa patients.

In 2022, Li and coworkers identified 27 metabolites differently expressed among BCa patients and cancer-free controls [72]. From these, adenosine was found with a decreased content in BCa cases, which is in agreement with the results obtained in the previously mentioned study of Wittman and colleagues [23]. Adenosine is a purine nucleoside with immunosuppressive properties in the tumor microenvironment; therefore, it is being reviewed as a potential target therapy in the field of oncology [73,74]. However, its involvement in the context of BCa cancer cells is poorly studied. Lastly, in the concern of discriminating between BCa patients and healthy controls, a study using a GC-MS-based approach discovered 24 volatile biomarkers for BCa detection [36]. Overall, these metabolites provided information related to metabolic dysregulations that might occur in BCa tumorigenesis, namely increased levels of alkanes and aromatic compounds and decreased levels of aldehydes. Among the increased levels of alkanes, only the 2,4-dimethylheptane and 4-methyloctane metabolites have been previously reported in association with cancer, being significantly increased in lung cancer cell lines [75], which suggests a cellular origin. With respect to the increased levels of aromatic compounds in urine samples, methylnaphthalenes have long been associated with chronic toxicity and carcinogenesis, especially in the case of smokers [76,77]. Therefore, the increased urinary levels of methylnaphthalenes may be related to tobacco use, which comprises one of the main risk factors for BCa development. Additionally, p-cresol is considered a uremic toxin that promotes the migration and invasion of carcinoma cells [78]. Thus, the increased content of p-cresol in urine samples of BCa patients may be related to the development of urothelial carcinoma.

Furthermore, some studies focused on BCa staging and grading. In that sense, in addition to the discrimination among BCa patients and healthy controls, Pinto et al. [36] found distinct urinary volatile profiles among patients diagnosed at different tumor stages. Some metabolites were similar to the ones noticed in the discrimination between BCa samples and control groups. However, octanal exhibited higher levels in tumor stage T1 in comparison to stage Ta/Tis, and methylglyoxal had a lower content between stage T2 and stage T1 of BCa. Octanal is a medium-chain aldehyde that has been reported in breast cancer cells and in the exhaled breath of lung cancer patients [79,80] but not in the case of BCa. Methylglyoxal can be formed as a byproduct of glycolysis and amino acid degradation. It is considered a highly reactive metabolite and is implicated in age-related chronic inflammatory diseases [81]. Notably, this metabolite has been demonstrated to promote cancer development via the Warburg effect and glycation [82]. Therefore, the increased content of methylglyoxal in urine samples between stage T2 and stage T1 of BCa patients may be related to the progressive extent of the malignancy. In addition, there were two metabolites, namely 1,2,3-trimethylbenzene and 1,2,4,5-tetramethylbenzene, which were increased in samples derived from stage T2 tumors when compared to stage Ta/Tis. [36]. Therefore, a higher extent of the disease may be related to an increased content of aromatic compounds that contain a monocyclic ring system consisting of benzene. In 2018, to address the differentiation between high-grade BCa (HG-BCa) and low-grade BCa (LG-BCa) for accurate grading, a study identified 58 altered metabolites between 15 HG-BCa and 18 LG-BCa cases [30]. These metabolites belonged to the pathways of histidine metabolism, retinol metabolism, arachidonic metabolism and tryptophan metabolism. Moreover, with the intent to discover diagnostic and staging biomarkers for BCa, a study recruited 198 BCa patients and 98 healthy volunteers and identified 15 dysregulated metabolites among different stages of NMIBC, MIBC and controls [18]. From these, urine p-cresol glucuronide was a downregulated metabolite and validated as a potential diagnostic biomarker for BCa, as well as a staging biomarker for NMIBC patients. This glucuronide derivate of the end-product of tyrosine biotransformation is generated by anaerobic intestinal bacteria typically excreted in urine [83,84]. Therefore, a reduction in urinary levels of this metabolite may indicate changes in the composition of the gut microbiota. In addition, spermine was a metabolite with increased levels, being validated as a staging biomarker for MIBC. This polyamine can also reflect changes in microbiota composition, contributing to the establishment of a tumor microenvironment that facilitates the initiation and progression of cancer, as previously reported [85].

Overall, the comprehensive analysis of urine metabolome reveals a set of metabolites that hold promising diagnosis value, distinctly characterizing various stages of BCa. These metabolites, whose urinary levels are modulated by bladder tumorigenesis, not only reflect systemic adaptations to cancer, such as inflammation, but also offer insights into tumor-specific metabolic changes.

#### 3.2.2. Data from Studies Performed with Blood-Derived Samples

Blood-derived fluids are commonly used in BCa research. The eight selected experimental papers that included blood-derived fluids’ metabolomics analysis gave rise to a list of 151 altered metabolites associated with BCa detection, grading and staging in blood-derived samples [Appendix A].

In 2012, Cao et al. performed an NMR-based metabolomic analysis of serum samples from 37 BCa patients to evaluate altered metabolic pathways related to BCa pathogenesis. They found significant changes in 10 metabolites between BCa patients and healthy patients [45]. Among them, the levels of certain intermediates and products of amino acids and glucose metabolism were decreased, while the levels of lipids and ketone bodies were increased. The increased content of lipids and ketone bodies is suggestive of enhanced lipogenesis, which may be related to the growth and development of tumoral tissue given that lipids are essential components of the basic structure of cell membranes [86]. Moreover, the decreased lactate levels observed in this study are contrary to the “Warburg effect” typically associated with tumorigenesis [87] but may be due to increased liver gluconeogenesis. A distinct NMR-based serum metabolomic analysis of 67 BCa patients also demonstrated altered levels of lactate compared to 32 healthy controls [88]; however, in this case, lactate was upregulated. Particularly, in this study, lactate was considered as a potential biomarker for the discrimination between BCa patients and healthy controls but was not sufficiently accurate to segregate LG- from HG-BCa, suggesting that it might not be associated with tumor aggressiveness [89].

Regarding the MS-based metabolomic approaches of blood-derived samples, a number of studies discriminate the global serum profile of BCa patients in terms of this malignancy’s detection and classification. A study by Lin et al. discriminated the global serum profile of 20 BCa patients, identifying five specific biomarkers for BCa detection [32]. These biomarkers included increased metabolites, namely eicosatrienol and docasotrienol, which are related to the metabolism of polyunsaturated fatty acids (PUFA), and azaprostanoic acid, which is an inhibitor of platelet aggregation [90]. The mechanism of platelet aggregation has been associated with cancer metastasis and progression [91], so the rise in azaprostanoic acid may be related to an immune response against BCa progression. In a different study, Nizioł et al. performed a serum metabolomics analysis of 100 BCa patients and 100 healthy controls. As a result, the molecular signatures revealed 27 metabolites that discriminated between BCa patients and healthy controls, 23 metabolites that differentiated between LG- and HG-BCa from controls and 39 metabolites that could distinguish BCa in distinct stages [33]. These metabolites were essentially lipids and lipid-like molecules. The structural diversity of lipids is reflected in distinct biological functions, including components of cell membranes, energy storage and cell signaling. Notably, cancer has been associated with alterations in lipid metabolism [92]. These alterations mainly include altered levels of glycerophospholipids, such as PCs, PSs and PEs, along with alterations in FA metabolism. As a result, cancer cells comprise distinct capabilities when compared to normal cells, for instance, a higher proliferation rate and higher resistance to chemotherapy [93,94]. Therefore, the observed altered lipid levels may reveal the adaptation of cancer cells underlying BCa development and progression, along with a harder BCa management for the patients included in the previously mentioned study. Interestingly, thymol has revealed a decreased content in BCa serum-derived samples in comparison to serum samples from healthy controls. It has been reported that this phenol is associated with anti-inflammatory effects, as well as with the inhibition of BCa cell proliferation [95]. Therefore, these results are in agreement, reinforcing the role of thymol as a potential therapeutic approach in BCa management.

In terms of BCa grading, an experimental study used LC-MS metabolomics of 120 BCa patients and identified 25 altered metabolites, from which they highlighted a promising three metabolites’ signature to classify and grade BCa [19]. This signature included higher levels of inosine, acetyl-N-formyl-5-methoxykynurenamine (AFMK) and PS (O-18:0/0:0) in HG-BCa vs. LG-BCa. Inosine is a nucleoside involved in purine metabolism. Its increased content seems to be associated with a more advanced state of the disease, namely HG-BCa, which is in agreement with the involvement of impaired purine metabolism in cell growth and progression of distinct cancer types [96,97]. On the other hand, AFMK has not been associated with cancer progression or development. Being a melatonin metabolite, it is mostly studied as a potential anti-inflammatory agent [98]. Therefore, the significance of its increased content in serum-derived samples from HG-BCa patients requires further investigation. In the case of PS (O-18:0/0:0), the higher levels of this phosphatidylserine (PS) in HG-BCa patients in comparison to LG-BCa are suggestive of impaired phospholipid metabolism. This glycerophospholipid is typically located in inner cellular membranes, and its exposure has been linked to immune suppression in the tumor microenvironment, being investigated as a potential target for cancer therapy [99]. Therefore, the increased PS content in serum samples of HG-BCa could indicate an impaired immune response to proliferating tumor cells. Another study also aimed to better categorize BCa patients, in this case using a GC-MS-based metabolomic approach for serum. This study identified 37 metabolites that were altered in HG-BCa patients compared to the control group, and only 16 metabolites differed between LG-BCa and the control group [20]. In general, there were significant increases in metabolite levels involved in FA and nucleotide synthesis and the pentose phosphate pathway in patients with HG-BCa. Particularly among the altered metabolites in the LG-BCa group, hippuric acid demonstrated a decreased content when compared to healthy controls. This is in agreement with the results of Tan and coworkers, in which hippuric acid content was equally decreased in BCa patients [19]. This metabolite is involved in phenylalanine metabolism. Nevertheless, it has been scarcely studied in connection with the pathophysiology of BCa, which is why its involvement in tumor development is unclear.

In general, comprehensive analysis of the blood metabolome reveals a number of metabolites that may play a role in the detection and progression of BCa. The variations in the concentrations of these metabolites in blood samples reflect some of the biological processes underlying BCa development. In addition, some of the tumor-specific metabolic changes reveal the immune response as well as structural changes in lipid content that may be related to the adaptation of BCa cancer cells.

#### 3.2.3. Data from In Vitro Studies

Regarding studies performed with cell lines using metabolomics-based approaches in the set of BCa, the selected papers highlighted 74 altered metabolites related to BCa diagnosis, classification and resistance to cisplatin-based treatment [Appendix A].

The study of Rodrigues et al. focused on the determination of a volatile metabolomic signature of BCa cell lines by a GC-MS approach. Initially, they used the BCa cell lines J82, Scaber and 5637 as well as the normal bladder cell line SV-HUC-1. It is important to note that each BCa cell line represents a different tumor grade and histological subtype. The BCa cell lines J82 and Scaber represent high-grade tumors of stage T3 and T4, respectively, while the BCa cell line 5637 represents a low-grade tumor at an unspecified stage. In addition, the BCa cell lines Scaber and 5637 represent the histologic subtype TCC, while the BCa cell line J82 is representative of the histologic subtype SCC. Analysis of the volatile metabolome revealed a panel of altered metabolites that differentiate between tumor and normal bladder cells. The major commonly altered metabolites included increased alkanes and ketones and decreased alcohols in all BCa cell lines compared to the normal bladder cell line [35]. These may be related to either FAO or FA synthesis for energy production or membrane formation, respectively, as well as inflammation processes and oxidative stress. In the matter of cancer pathophysiology, these alterations reflect how BCa cancer cells make use of FA metabolism for their growth and proliferation, which corroborates previous studies [100,101,102]. FAs are essential components of cell membranes and important sources of energy. Therefore, during cancer development and proliferation, they can be used either for membrane formation or as an energy source. In fact, it is known that cancer cells can reprogram their metabolism to meet higher energy demands for higher replication rates and rapid proliferation. Furthermore, tumor-associated macrophages have been shown to use FAO for energy production, leading to the production of ROS, which are a hallmark of cancer [100,103]. Interestingly, the metabolite 2-pentadecanone, a ketone, was altered in all BCa cell lines in this study. In this type of cancer, this ketone has been demonstrated to be derived from a metabolic cascade starting from the cellular metabolism of glucose to FA synthesis [104], which, as stated before, are metabolic pathways that are typically abnormal in cancer cells. Though, it exhibited distinct regulation trends, it was upregulated, except in the Scaber BCa cell line. Hence, its regulation may be influenced according to the histological subtype of BCa. Nonetheless, it has already been reported as an altered metabolite in gastric cancer cell lines [105]. To create a volatile signature to assess tumor aggressiveness, this research group also compared low-grade (5637) and high-grade (Scaber and J82) cell lines, resulting in 21 significantly altered volatile organic compounds. Most of them showed lower levels in HG-BCa compared to LG-BCa, although benzaldehyde showed increased levels [35]. These results may reflect the energetic demands associated with a higher grade of malignancy. For instance, lipid peroxidation of cellular membranes for energy production may result in the production of aldehydes. At last, they compared TCC (J82 and 5637) and SCC (Scaber) cancer cell lines, which resulted in a panel of 14 altered metabolites demonstrating that distinct histological subtypes have different volatile organic compounds (VOCs).

Later, in 2019, the same research group analyzed the metabolome of the LG-BCa cell line 5637 and HG-BCa cell line J82 to find altered metabolites and assess tumor aggressiveness by GC-MS. Metabolomic analyses revealed altered levels of amino acids and FAs in HG-BCa compared to LG-BCa, of which glycine, myristic acid, palmitic acid and palmitoleic acid were downregulated, while aspartic acid, leucine, methionine and valine were upregulated [47]. This result shows that BCa cells from different classes have different metabolic profiles in terms of amino acid metabolism and FA biosynthesis to compensate for the higher energetic requirements. That is, to meet the high energy requirements for growth and proliferation, cancer cells can metabolize FAs that provide large amounts of energy for cellular processes. In terms of amino acid metabolism, the reduced amino acid content could be related to increased protein synthesis for cancer cell proliferation.

Remarkably, Lee and coworkers attempted to characterize the lipidomic profile associated with cisplatin resistance in BCa cells using an LC-MS approach. For this purpose, they used two isogenic human T24 BCa cell lines: one was cisplatin-sensitive (T24S) and the other was cisplatin-resistant (T24R). Accordingly, they found 16 differentially expressed lipids that are most likely associated with cisplatin resistance and tumor aggressiveness, suggesting potential lipid species as biomarkers for identifying higher-risk patients [94], such as sphingomyelin (SM), ceramide (CE), phosphatidylcholine (PC), phosphatidylethanolamine (PE) and triglycerides (TGs). PC and PE are two of the most abundant glycerophospholipids in cell membranes. Therefore, their increase can be associated with membrane formation. However, an increase in PC and a decrease in PE were found to be associated with cisplatin-resistant cells, suggesting that higher PC content may be an indicator of poorer cancer prognosis in BCa patients. In addition, decreased TG levels were found in this study, which in turn may be associated with the higher energy requirements of a more aggressive stage of disease, i.e., increased FA metabolism in BCa cells.

Overall, the comprehensive analysis from in vitro studies with different BCa cell lines revealed a number of metabolites that are altered in response to BCa development. These metabolites mainly include amino acids and lipids or lipid-like molecules and shed light on metabolic changes that may occur during tumorigenesis in the bladder to sustain cancer cell growth and proliferation, as well as structural changes in cancer cell membranes that are associated with BCa pathogenesis.

#### 3.2.4. Data from Studies with Ex Vivo Bladder Tissue

Bladder tissue-derived samples require the most invasive approach in the context of BCa research, explaining at least in part the low number of experimental studies comprising this type of sample. Notwithstanding, the selected studies provided a list of 1031 altered metabolites related to BCa diagnosis and staging with either human or mice tissue-derived samples [Appendix A].

In the study of Vantaku et al., they used high-resolution LC-MS combined with bioinformatic analysis for the determination of the global metabolome and lipidome of 25 BCa tissue-derived samples. Consequently, 533 metabolites were found to be altered between low- and high-grade BCa. In general, the altered metabolites included nucleotides, polyamines, prostaglandins and carnitines, among others, while the altered lipids comprised seven distinct lipid classes, specifically cardiolipin (CL), PC, PE, phosphatidylinositol (PI), plasmenyl-PE, PS and TG [38]. In addition, taking into account the identified differentially expressed metabolites and lipids, the metabolic pathway analysis performed in this study revealed the deregulation of glycerophospholipid and arachidonic acid metabolism [38]. Particularly, it has been demonstrated that there is a decrease in PC content in bladder tissue derived from LG-BCa patients when compared to tissue from HG-BCa patients. This result is in agreement with the findings reported by Lee and colleagues in which PC was increased in the case of higher-risk BCa patients [94]. Therefore, the possibility of higher levels of PC being related to a more aggressive state of BCa is reinforced, most likely due to the higher need for membrane formation required for increased cell proliferation. In addition, bladder tissue derived from LG-BCa patients has revealed higher levels of arachidonic acid, a PUFA typically associated with inflammation, which suggests an inflammatory state in BCa patients with a lower grade of the disease. Contrastingly, in a previously mentioned study, arachidonic acid exhibited a higher content in urine-derived samples from BCa patients who were mainly in a higher-grade state of malignancy [23]. These contradictory results emphasize the need for a deeper understanding of the role of arachidonic acid in BCa carcinogenesis. In 2021, Tu and colleagues performed infrared matrix-assisted laser desorption electrospray ionization MS imaging (IR-MALDESI MSI) of six MIBC and six adjacent matched normal tissue samples. As a result, they identified 408 altered metabolites that allowed differentiation between tumor and normal tissue [106]. The vast majority of these metabolites included overexpressed PC, PE, PI and PA lipid species, along with glycerides and free FAs. In contrast, normal tissue samples had a greater number of SM and cholesterol ester species [106]. Notably, the MIBC specimens were classified in HG-BCa of stages T2, T3 and T4 according to the 2016 WHO classification [107]. Taking this into consideration and comparing the results between the two previously mentioned studies, four common metabolites were identified, namely bolasterone, PE (32:1), PC (36:2) and PC (36:3) [38,106]. Therefore, these metabolites could be potential biomarkers for the diagnosis of high-risk patients with BCa.

As mentioned above, the heterogeneity of BCa can be divided into several different molecular subtypes. Therefore, the study by Feng et al. focused on distinguishing between basal and luminal subtypes of MIBC. To achieve this goal, they used an LC-MS-based approach to identify and quantify metabolites in 12 tissue samples of MIBC. Their results revealed 73 altered metabolites in the basal and luminal MIBC subtypes, suggesting that sulfatides (SLs), SMs and free FAs mainly contribute to the diagnosis of the basal MIBC subtype, while PCs and PEs are typically more abundant in the luminal subtype [39]. Thus, these metabolites may be useful to provide an improved differential diagnosis of MIBC.

In addition to human tissue samples, bladder tissue was obtained from a mouse disease model system for the investigation of BCa. In the study by Ho et al., for example, a BCa mouse model was used that was produced by the instillation of an MB49 mouse bladder carcinoma cell line. They then examined the composition of the lipid extracts obtained from the bladder tissue of 10 tumor-bearing and 10 healthy mice using an LC-MS-based method. The comparative analysis revealed that phospholipids enriched in unsaturated acyl groups, such as PC (36:4) and PC (38:4), and SM, such as SM (36:1), were the most abundant lipid species in tumors compared to healthy samples [40]. More so, ceramide with long-chain saturated FA parts revealed a lower abundance in tumor-derived tissue when compared to healthy tissue [40]. In addition, one of the most abundant lipid species found in tumor tissue-derived samples was PE and, more precisely, PE (32:1). Notably, this lipid was equally found in the previously mentioned experimental studies regarding tissue-derived samples, in which it was upregulated in HG-BCa-related studies and downregulated in the comparison between distinct MIBC molecular subtypes [38,39,106]. This suggests that this phospholipid may be a potential biomarker in BCa detection, especially in the high-grade state of the disease, excluding the basal molecular subtype.

### 3.3. Integrative Analysis of Data from Studies Performed with Distinct Types of Samples

Blood- and urine-derived samples are the most common human fluids used in BCa research. Although urine samples are obtained by a less invasive approach compared to blood samples, they equally reflect the metabolic changes and molecular processes that can occur during BCa development. In a study by Yu and coworkers, the serum and urine metabolome of 26 BCa patients, 15 of whom had NMIBC and 11 of whom had MIBC, was determined using an LC-MS method. As a result, they found 13 serum and 13 urine metabolites that were significantly different in NMIBC compared to MIBC samples. In addition, the mechanistic mapping performed by this research group revealed that the discovered metabolomic signature is related to immune, metabolic and inflammatory responses [22]. However, only one of the metabolites found, pelargonic acid, was present in serum and urine samples, suggesting that these human fluids may contain different molecular information on BCa progression and should be analyzed separately. More importantly, this study again found that of the metabolome profiles found, only the overall urine metabolome profile was significantly different in NMIBC and MIBC, suggesting that urine metabolomics may be a preferred approach for the discovery of potential biomarkers for the study of BCa progression [22].

With this in mind and in the context of the present review, an integrative analysis of the data collected from the selected papers using either blood- or urine-derived samples was performed [Appendix A]. This integrative analysis revealed a total of 17 common metabolites, as shown in Figure 3. Most of these metabolites showed a consistent trend in terms of increased or decreased levels, with a few exceptions, e.g., lactate, choline, glycine and decanoylcarnitine. Interestingly, the common metabolites of blood- and urine-derived samples, which show the same trend in all studies, are consistent with previous conclusions on possible adaptations that may occur during BCa development, i.e., impaired FA metabolism. The increased level of pelargonic acid, a fatty acid, could be related to the previously mentioned increase in FA synthesis for membrane formation during cancer cell growth and proliferation. In addition, an increase in the content of acylcarnitines (L-acetylcarnitine and 9-decenoylcarnitine) is also highlighted, which may be associated with energy production by FAO due to the higher energy requirements of cancer cells. Nevertheless, the common metabolites account for only a small fraction of the global metabolome signatures found in a total of 18 studies. This once again underlines the fact that the blood and urine metabolome reflect different adaptations of the body to BCa development, e.g., in the case of lactate content in blood and urine samples. Therefore, complementary information can be obtained from the metabolomic analysis of both fluids. Urine seems to be a better sample for the identification of metabolites released by the bladder tumor and thus for the identification of molecular processes up- and downregulated in the tumor.

Regarding the use of human fluids and ex vivo bladder tissue in BCa omics research, integrative analysis of the 22 selected papers that included these types of samples revealed a total of 25 common metabolites between studies (overview in Figure 4). The vast majority of the potential impact of some of these metabolites on BCa development and progression has already been discussed in the present systematic review, particularly in relation to lipids and lipid-like molecules, including FAs (such as linoleic acid, arachidonic acid and myristic acid), suggesting impaired lipid metabolism during BCa development and progression. In addition, oleamide was found to have elevated urinary levels in BCa patients associated with high tumor malignancy and T1 and T2 stage of cancer [34], as well as in bladder tissue samples from HG-BCa patients [106]. Oleamide is an endogenous lipid that has previously been demonstrated to exert an effect on Ca^2+^ signaling in human BCa cells; that is, it led to an increase in the intracellular content of Ca^2+^ in a phospholipase C-independent way [108]. However, there are no recent studies on its role in the development of this malignant disease. Therefore, its role in BCa pathology needs to be further investigated. In the integrative analysis of human fluid and bladder tissue samples derived from BCa patients, it is important to draw attention to the fact that these common metabolites are most abundant in urine and bladder tissue samples, supporting the idea that urine better reflects the molecular changes underlying tumor development and is preferred for the identification of potential biomarkers for the early BCa diagnosis and the follow-up of cancer management (Appendix A).

In terms of the use of human fluids vs. cell lines as samples in BCa research, within the 21 selected papers regarding the use of blood, urine and cell line cultures for BCa research, seven common metabolites were found (Figure 5). These include branched-chain amino acids (BCAA), valine and leucine [23,47] as increased metabolites, along with glycine [45,47] and myristic acid [17,47] as decreased metabolites. Notwithstanding, in the case of valine, leucine and glycine, these have recently displayed contradicting results in the study of Ossoliński and coworkers; that is, valine and leucine exhibited decreased urinary levels, whereas glycine revealed increased urinary levels in comparison to healthy controls [46]. Interestingly, BCAA metabolism has been marked as a potential oncogenic metabolic pathway in the sense that it may be perturbed due to increased biosynthetic and nutritional demands involved in tumor progression [109]. Furthermore, glycine is a proteogenic amino acid also linked to tumorigenesis, which is being reported as an important fuel for cancer cell proliferation [110,111]; therefore, its increased consumption can lead to a lower content in cancer cells. The modulation of these amino acids in the context of BCa pathology therefore requires further investigation. In addition, both myristic and palmitic acids showed the same trend of lower levels in urine samples and BCa cells in all studies. This finding highlights the involvement of FA in BCa pathology, especially in the context of the role of FAO as an energy source for tumor development and progression. Regarding the type of samples used in BCa research, human fluids appear to have similar molecular expression compared to cell line cultures.

## 4. Conclusions

A total of 1562 metabolites were reported as altered in 25 experimental studies on molecular alterations associated with BCa in terms of development, detection and prognosis. Within the different sample types, most of the reported metabolites were found altered in the bladder tissue of BCa patients, which is consistent with the fact that this type of sample represents a direct approach to studying the tumor and its microenvironment. However, it is also the sample type with the lowest number of studies, as sample collection must be invasive. In addition, both blood- and urine-derived samples can be representative of the molecular changes underlying BCa. Among these two fluids, urine appears to display a higher likelihood of containing metabolites that are also present in ex vivo bladder tissue and cell line cultures. In this comparative analysis of different sample types, notable variations in several metabolites were consistently identified. Specifically, elevated levels of L-isoleucine, L-carnitine, oleamide, palmitide, arachidonic acid and glycoursodeoxycholic acid, along with decreased levels of deoxycytidine, 5-aminolevulinic acid and pantothenic acid were observed in both bladder tissue and urine samples. These findings warrant consideration as potential components of a BCa metabolome signature. Indeed, one of the main limitations in the literature is the scarcity of metabolomic studies conducted with ex vivo bladder tissue from BCa patients, as well as reference groups for comparative analysis of metabolite level variations. Some studies compare them with healthy subjects, while others focus on those with low-grade BCa (Appendix A). Remarkably, no specific biological processes appear to predominate, as shown by the lack of associations between these metabolites, as depicted in the MetScape analysis. Furthermore, only one metabolite from each pathway was consistently found in urine and BCa tissue or cells (Figure 6). Vitamin B5-CoA biosynthesis from pantothenate and porphyrin metabolism was found to be downregulated, considering the metabolites involved, whereas arachidonic acid metabolism, pyrimidine metabolism, BCAA metabolism and fatty acid metabolism are upregulated. Nevertheless, CoA, derived from pantothenate, assumes a central role in bridging these pathways by acting as a cofactor for crucial enzymatic reactions involved in fatty acid metabolism, amino acid metabolism and porphyrin metabolism. However, the downregulation of CoA biosynthesis seems to correlate with the downregulation of porphyrin metabolism but with the upregulation of fatty acid metabolism, particularly arachidonic acid, and BCAA metabolism, suggesting the involvement of complex regulatory signaling pathways.

Notwithstanding, the use of metabolomics emerges as an essential tool enabling the identification of metabolite signatures for a non-invasive early diagnosis, assessment of recurrence and progression and monitoring of therapeutic responses within distinct BCa subtypes. However, these identified metabolites should be validated in future studies, taking into account potential influencing factors such as age, gender, stage and grade of tumor and other risk factors associated with the development of BCa.

The future of Omics-based research in BCa is promising. Nevertheless, there are still some challenges to overcome in order to achieve better development and application of this research area in the different fields of oncology. For example, a standardized experimental design and methodological approach should be introduced to achieve more comparable results between studies. Further studies should enable the creation of a database of all metabolite characteristics and their interactions to better understand the biological processes underlying BCa development. Finally, the development of high-throughput analytical platforms in combination with bioinformatics tools could enable a multi-Omics approach to capture all cancer features. Taking these aspects into account, it may be possible to achieve the main goal of using Omics-based approaches in BCa research: early diagnosis through noninvasive approaches, better stratification of cancer patients, prediction of recurrence risk and an improved therapeutic follow-up while increasing the knowledge of tumor biology.

## Figures and Tables

**Figure 1 ijms-25-03347-f001:**
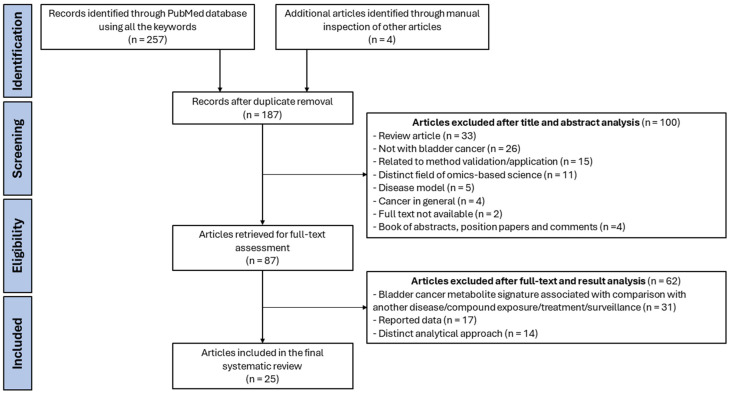
PRISMA flowchart providing a visual summary of the study selection process of the 25 experimental studies included for the systematic review of metabolomics-based research in bladder cancer.

**Figure 2 ijms-25-03347-f002:**
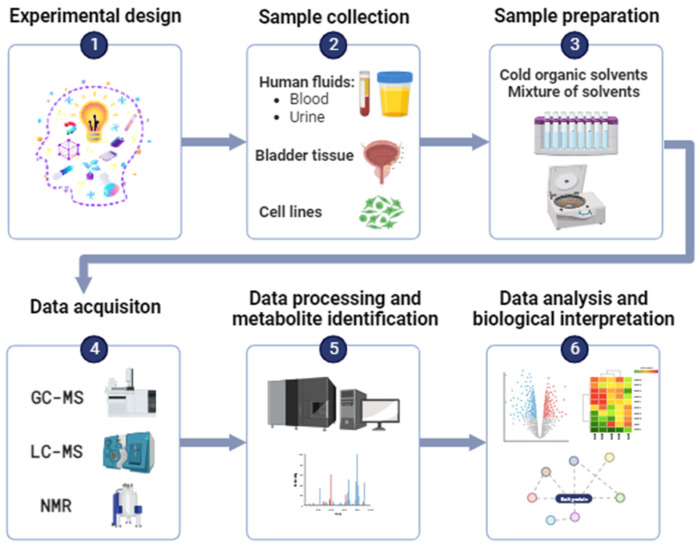
Overview of the methodological workflow used for a bladder cancer metabolomics-based approach using nuclear magnetic resonance (NMR) or mass spectrometry (MS) analytical platforms.

**Figure 3 ijms-25-03347-f003:**
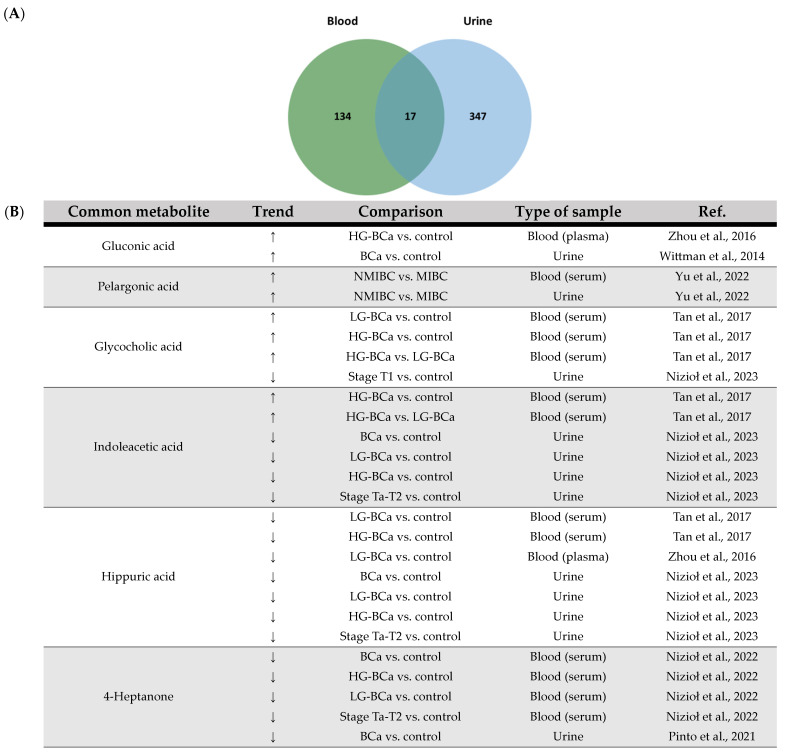
(**A**) Venn diagram of the integrative analysis of data from the 18 selected papers regarding bladder cancer using a metabolomic approach for the analysis of urine- and blood-derived samples. (**B**) Common metabolites between urine and blood and respective trend regulation by bladder cancer and type of sample. ↑: increase; ↓: decrease. BCa: bladder cancer; HG: high grade; LG: low grade; NMIBC: non-muscle-invasive bladder cancer; MIBC: muscle-invasive bladder cancer [17,19,20,22,23,30,31,33,34,36,45,46,54,72].

**Figure 4 ijms-25-03347-f004:**
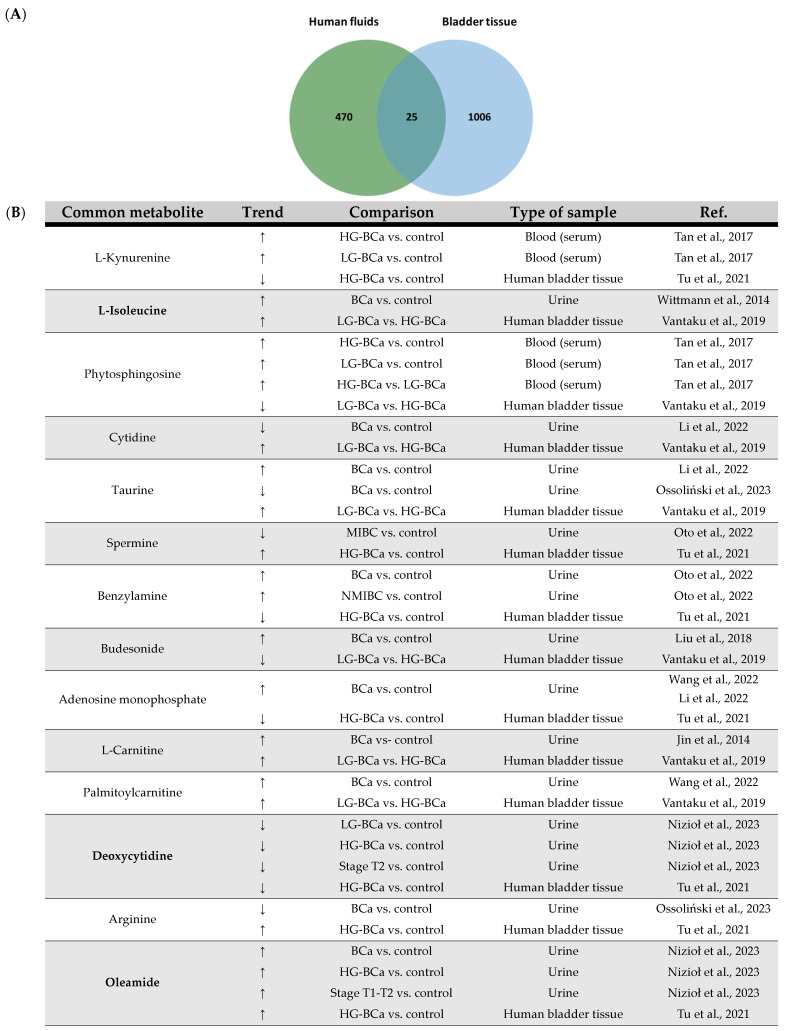
(**A**) Venn diagram of the integrative analysis of data from the 22 selected papers regarding bladder cancer using a metabolomic approach with human fluid and ex vivo bladder tissue samples. (**B**) Common metabolites between human fluids and ex vivo bladder tissue and respective trend regulation by bladder cancer and type of sample. Metabolites consistently modulated by BCa in both sample types are highlighted in bold, while those exhibiting a distinct trend in cell lines are marked in bold blue. ↑: increase; ↓: decrease. BCa: bladder cancer; HG: high grade; LG: low grade; NMIBC: non-muscle-invasive bladder cancer; MIBC: muscle-invasive bladder cancer [17,18,19,23,30,33,34,38,39,46,54,60,72,106].

**Figure 5 ijms-25-03347-f005:**
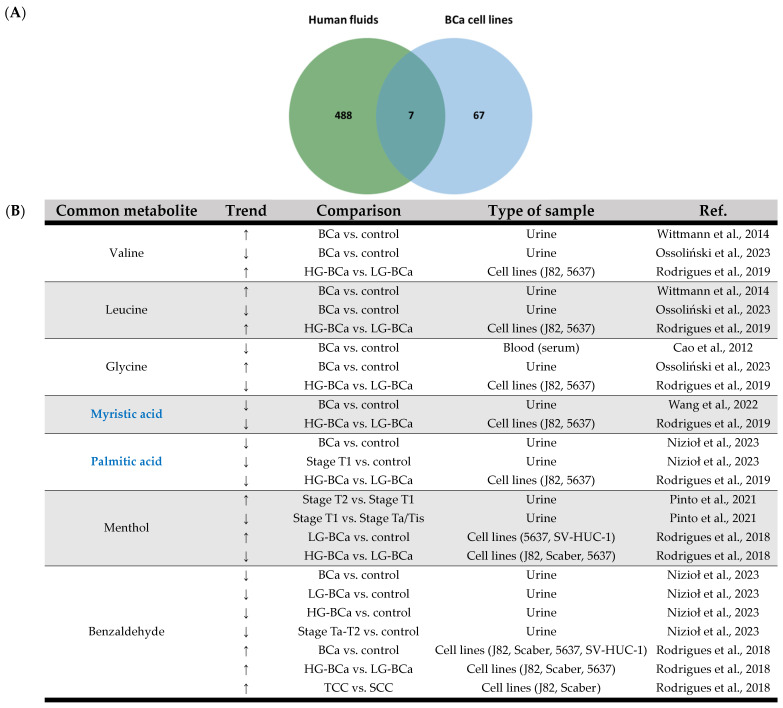
(**A**) Venn diagram of the integrative analysis of data from the 21 selected papers regarding bladder cancer using a metabolomic approach with human fluid and cell line culture samples. (**B**) Common metabolites between human fluids and cell lines and respective trend regulation by bladder cancer and type of sample. Metabolites exhibiting a distinct trend in bladder tissue are marked in bold blue. ↑: increase; ↓: decrease. BCa: bladder cancer; HG: high grade; LG: low grade; TCC: transitional cell carcinoma; SCC: squamous cell carcinoma [17,23,34,35,36,45,46,47].

**Figure 6 ijms-25-03347-f006:**
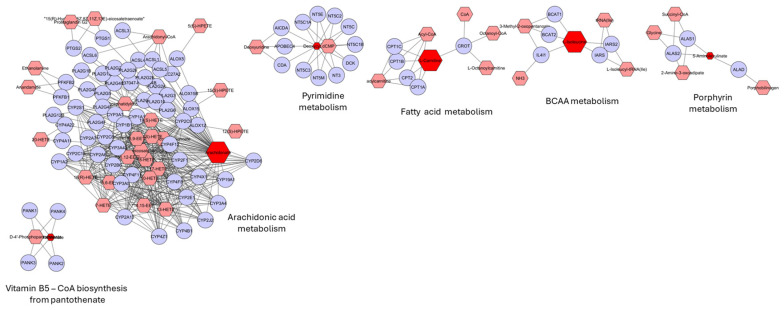
MetScape (v3.1.3; https://metascape.org/) analysis used to examine metabolites that are equally modulated by bladder cancer in both urine and bladder cancer tissue, highlighting the pathways in which these metabolites are actively involved. Enlarged red nodes correspond to metabolites found at higher levels, whereas smaller red nodes represent those present at lower levels in BCa settings. The metabolites from which the identified compounds may originate or derive are represented by pink hexagons, while the proteins (identified by gene name) involved in the metabolic processes are denoted by blue circles. The comprehensive details of pathway enrichment, including the associated metabolites for each pathway along with their corresponding FDR and *p*-values, are presented in Appendix A.

## Data Availability

Data are contained within the article or Appendix A.

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
