# Peer review of "Unmasking the Metabolite Signature of Bladder Cancer: A Systematic Review"

_ijms, 2024, doi:10.3390/ijms25063347_

Round 1
Reviewer 1 Report
Comments and Suggestions for Authors
The work by Pereira et al. reviews 25 experimental studies that all applied metabolomics analyses in the context of bladder cancer. The article first introduces to the significance of the disease and its pathological features, and continues with a summary of the main analytical techniques used in metabolomics of bladder cancer. It then lists in detail the metabolic changes found in the different studies for urine, blood, in vitro cell culture and ex vivo tissue samples, and concludes with a meta-analysis of metabolite changes common to more than one of the sample types. The article is generally well researched and written and of interest to the readers of IJMS. A carefully compiled metabolite list is included as Supplement that can be very valuable for further data integration.
Major comments:
(1) The chapters reviewing the individual studies (2.2.1 to 2.2.4) are rather lengthy compared to the integrative chapter (2.3). Wouldn't it make more sense to discuss in detail thoses metabolic changes that are commonly found by two or more studies? Please comment on why the present distribution was chosen.
(2) Similar review/meta-analysis articles on bladder cancer metabolomics were already published, including one in IJMS in 2022:
di Meo et al. 2022, https://doi.org/10.3390/ijms23084173
Garcia-Perdomo et al. 2023, https://doi.org/10.1016/j.ajur.2022.11.005
Cheng et al. 2015, https://pubmed.ncbi.nlm.nih.gov/26379905
Zhang et al. 2018, https://doi.org/10.2217/bmm-2018-0229
Amara et al. 2018, https://doi.org/10.1080/14789450.2019.1583105
Unfortuneatly, none of these references are cited. Please include these references and point out the differences to your work.
(3) It remains unclear how the 'final' set of metabolites (l642, l25) was chosen. Were these the ones most frequently found in the different studies? Or most frequently found to have the same trend? This should be stated precisely.
(4) Different Venn diagrams are shown for two sample types at a time (Figs. 2-4). A 4-way Venn diagram integrating all four sample types would be highly welcome and could possibly underpin the 'final' biomarker recommendation.
(5) The tables in Figs. 2-4 could be ranked by number of evidences per metabolite. It is unclear what the metabolites are sorted by at present.
Minor comments:
l24: "Urine ... more likely to have": unclear what 'more' refers to
l75: "present review": it is actually also a meta-analysis - why not call it like this?
l223: replace "it was observed a decreased" - check language
Chapter 3:
- How was the overlap of metabolites between studies determined? Based on metabolite names, IUPAC names, InChIKey's? This should be specified.
- How were chemical names normalized? Which name was chosen as default? PubChem names?
- Metabolite meta data in Tables S2 and following are incomplete - please complete where possible.
- "1567 altered metabolites": table S2 lists 1562 unique metabolites - please check.
- Fig. S2 not explained: how was data prepared for MetScape, what parameter/database settings were used, what was MetScape version etc.
l640: "containing a greater number of metabolites": check sentence (you mean "more significant changes in metabolite abundances")
Fig. S1: "PRISMA" needs to be explained/cited
Comments on the Quality of English Language
The language is generally adequate but should checked before publication. I mentioned a few inconsistencies in the comments.
Author Response
Reviewer #1
The work by Pereira et al. reviews 25 experimental studies that all applied metabolomics analyses in the context of bladder cancer. The article first introduces to the significance of the disease and its pathological features, and continues with a summary of the main analytical techniques used in metabolomics of bladder cancer. It then lists in detail the metabolic changes found in the different studies for urine, blood, in vitro cell culture and ex vivo tissue samples, and concludes with a meta-analysis of metabolite changes common to more than one of the sample types. The article is generally well researched and written and of interest to the readers of IJMS. A carefully compiled metabolite list is included as Supplement that can be very valuable for further data integration.
Major comments:
(1) The chapters reviewing the individual studies (2.2.1 to 2.2.4) are rather lengthy compared to the integrative chapter (2.3). Wouldn't it make more sense to discuss in detail those metabolic changes that are commonly found by two or more studies? Please comment on why the present distribution was chosen.
R: We appreciate the Reviewer's viewpoint; nevertheless, our intent was to comprehensively address the metabolites modulated in BCa at the first time of their mention in the manuscript. Then, in the integrative section (section 2.3.), we conducted a more thorough exploration into the origin of these metabolites and delved into their potential biological significance within the context of BCa.
(2) Similar review/meta-analysis articles on bladder cancer metabolomics were already published, including one in IJMS in 2022:
di Meo et al. 2022, https://doi.org/10.3390/ijms23084173
Garcia-Perdomo et al. 2023, https://doi.org/10.1016/j.ajur.2022.11.005
Cheng et al. 2015, https://pubmed.ncbi.nlm.nih.gov/26379905
Zhang et al. 2018, https://doi.org/10.2217/bmm-2018-0229
Amara et al. 2018, https://doi.org/10.1080/14789450.2019.1583105
Unfortuneatly, none of these references are cited. Please include these references and point out the differences to your work.
R: Acknowledging the Reviewer's feedback, we have made amendments in the Introduction section of the manuscript. Specifically, we referenced the reviews highlighted by the reviewer that have previously analysed the literature on metabolomics alterations in BCa, emphasizing the distinctiveness of our review. Therefore, in the revised version of the manuscript, it reads as follows: “Moreover, efforts have been made to integrate data from metabolomics studies to pro-pose biomarkers for BCa management [24–28]. In the present review, we aimed to delve deeper into the understanding of the molecular changes occurring during BCa development through an updated systematic literature search of experimental studies focusing on BCa metabolite profiling, followed by an integrative analysis using bioinformatic tools. Data from this analysis undergo critical discussion of the reported data. A list of altered metabolites associated with the detection, development, and prognosis of BCa that can be considered as potential biomarkers is presented.”
(3) It remains unclear how the 'final' set of metabolites (l642, l25) was chosen. Were these the ones most frequently found in the different studies? Or most frequently found to have the same trend? This should be stated precisely.
R: Addressing the Reviewer's concern, we have made clarifications in the revised version of the manuscript to elucidate the selection criteria for the nine metabolites mentioned in the abstract and conclusion sections. These particular metabolites exhibited consistent variations between urine and bladder tissue, with those demonstrating distinct variations in cell lines (specifically myristic acid and palmitic acid) being excluded. In the conclusion section of the revised manuscript, the following idea was expressed: “In this comparative analysis of different sample types, notable variations in several metabolites were consistently identified. Specifically, elevated levels of L-isoleucine, L-carnitine, oleamide, palmitide, arachidonic acid and glycoursodeoxycholic acid, along with decreased levels of deoxycytidine, 5-aminolevulinic acid, and pantothenic acid were observed in both bladder tissue and urine samples. These findings warrant consideration as potential components of a BCa metabolome signature.”
In Section 5 of the revised version of our manuscript, we have also incorporated the following idea to highlight the limitations of the integrative analysis performed in our study: “Indeed, one of the main limitations in the literature is the scarcity of metabolomic studies conducted with ex vivo bladder tissue from BCa patients, as well as reference groups for comparative analysis of metabolite level variations. Some studies compare with healthy subjects, while others focus on those with low-grade BCa (Supplementary Table S2).”
(4) Different Venn diagrams are shown for two sample types at a time (Figs. 2-4). A 4-way Venn diagram integrating all four sample types would be highly welcome and could possibly underpin the 'final' biomarker recommendation.
R: We understand the Reviewer's suggestion, and indeed, we considered creating a figure that present a 4-way Venn diagram comparing the altered metabolites in the different sample types. However, in the end, we chose to create a separate figure for each comparison because we included in each figure the list of common metabolites and how they varied in each sample. By consolidating all comparisons into one figure, we believed it would be challenging to track the variations of metabolites across different comparisons. However, in the revised manuscript, we have emphasized in blue and bold the metabolites that are shared with other samples (figure 4), aside from those directly under comparison as highlighted in figure 3, also emphasized in bold. This information was included in the legends of both figures.
(5) The tables in Figs. 2-4 could be ranked by number of evidences per metabolite. It is unclear what the metabolites are sorted by at present.
R: While we may not completely understand the Reviewer's concern, it's important to note that the metabolites listed in Figs 2-4 were chosen based on the Venn diagram, specifically focusing on those metabolites common to the samples under comparison. Then, we have considered the variation trends reported in each study for comparison (based on the information presented in Supplementary Table S2).
Minor comments:
l24: "Urine ... more likely to have": unclear what 'more' refers to
R: For enhanced clarity, we revised the sentence to read: “Urine samples displayed a higher likelihood of containing metabolites that are also present in bladder tumor tissue and cell line cultures.”
l75: "present review": it is actually also a meta-analysis - why not call it like this?
R: In fact, we conducted a systematic review of the literature, and we employed tools such as Venny and MetScape for the integrative analysis of the data retrieved from the literature. We did not utilize meta-analytical methods to synthesize the findings or generate a summary effect size or outcome. Therefore, we refrain from categorizing our review as a meta-analysis. Reviewer #2 similarly contends that this review does not qualify as a meta-analysis.
l223: replace "it was observed a decreased" - check language
R: The correction was made.
Chapter 3:
- How was the overlap of metabolites between studies determined? Based on metabolite names, IUPAC names, InChIKey's? This should be specified.
R: The identification of overlapping metabolites between studies was conducted utilizing the Venny tool, which relied on metabolite names for comparison. This information was included in Section 3 of the revised version of the manuscript.
- How were chemical names normalized? Which name was chosen as default? PubChem names?
R: The metabolite names as they appeared in the original papers were carefully considered and utilized for overlap analysis between studies. Additional metadata provided in Supplementary Table S2 was extracted from the original paper whenever feasible; however, when unavailable, it was sourced from PubChem and HMDB. To elucidate this process further, in Section 3 of the revised manuscript, we have incorporated the following idea: “The metabolite names and other metadata from the original paper were retained whenever possible. In cases where this information was not provided, a search was conducted in PubChem and HMDB.”
- Metabolite meta data in Tables S2 and following are incomplete - please complete where possible.
R: We acknowledge that there is missing information regarding certain metabolites. Indeed, the metabolite metadata listed in Supplementary Table S2 was provided for those that were available, primarily relying on the information from the original article. Alternatively, when such information was not available in the original paper, it was retrieved from PubChem and HMDB. However, for many metabolites, information is not available in these repositories.
- "1567 altered metabolites": table S2 lists 1562 unique metabolites - please check.
R: We appreciate the Reviewer's thorough examination of the article. In fact, we overlooked verifying the information regarding the number of metabolites. In the revised version of the article, we have corrected this to 1562 distinct metabolites.
- Fig. S2 not explained: how was data prepared for MetScape, what parameter/database settings were used, what was MetScape version etc.
R: Supplementary Figure S2 was created in MetScape, taking into account the 9 metabolites that exhibit consistent variations among urine, bladder tissue, and cell lines. Of these 9 metabolites, KEGG IDs were only available for 6 metabolites. MetScape utilizes these KEGG IDs for network analysis. Fold change variations (up or down) in bladder cancer samples were considered in this analysis, resulting in metabolite nodes with distinct sizes (enlarged when in higher levels or smaller when in lower levels). To enhance clarity regarding the analysis conducted, the following idea was included into Section 3 of the revised version of the manuscript: “For the metabolites consistently modulated by BCa in urine, bladder tissue, and cell lines, a MetScape analysis (v3.1.3.; http://metscape.ncibi.org) was conducted, and the findings are illustrated in Supplementary Figure S2. Out of the 9 metabolites identified as modulated by BCa, only six were included in this analysis due to the availability of their KEGG IDs. The size of the nodes reflects variations in the metabolite levels; they appear enlarged when at higher levels and smaller when at lower levels in samples from subjects with BCa.” Supplementary Figure S2 has been revised to incorporate the biological processes associated with the metabolites modulated by BCa. Additionally, the legend of Supplementary Figure S2 has been expanded for enhanced clarity.
l640: "containing a greater number of metabolites": check sentence (you mean "more significant changes in metabolite abundances")
R: The sentence was revised to: “Among these two fluids, urine appears to display a higher likelihood of containing metabolites that are also present in ex vivo bladder tissue and cell line cultures.”
Fig. S1: "PRISMA" needs to be explained/cited
R: Supplementary Figure S1 was referenced in section 3; however, in the revised version of the manuscript, we also cited this figure in line 118.
Reviewer 2 Report
Comments and Suggestions for Authors
The authors systematically analyzed the results of 25 published papers. They described the main methods of metabolomics, metabolites and pathways associated with bladder cancer from the mentioned works. The article is very useful and valuable. However, some aspects must be improved.
1) The article has the wrong type and format. This is not a paper of the "article" type, not even a meta-analysis, this is a "review". It is unethical to claim descriptions of previous studies as results. The format and type must be changed to the appropriate ones.
2) Some metabolic pathways have been mentioned as altered in bladder cancer. It is usually possible to estimate a small part of the pathway, since measuring different metabolites in a single profile is a difficult task. How many metabolites in these pathways have been evaluated (in absolute numbers and percentages)?
3) Statistical methods are not described in detail for the mentioned studies. Are they the same or comparable? What about preprocessing the metabolic data in the studies mentioned, e. g., was any uniform normalization there?
Author Response
The authors systematically analyzed the results of 25 published papers. They described the main methods of metabolomics, metabolites and pathways associated with bladder cancer from the mentioned works. The article is very useful and valuable. However, some aspects must be improved.
1) The article has the wrong type and format. This is not a paper of the "article" type, not even a meta-analysis, this is a "review". It is unethical to claim descriptions of previous studies as results. The format and type must be changed to the appropriate ones.
R: The Reviewer is entirely correct, and we appreciate the careful review of our article. Indeed, it was an oversight on our part that has been corrected in the revised version of the article.
2) Some metabolic pathways have been mentioned as altered in bladder cancer. It is usually possible to estimate a small part of the pathway, since measuring different metabolites in a single profile is a difficult task. How many metabolites in these pathways have been evaluated (in absolute numbers and percentages)?
R: We acknowledge the Reviewer's comment, but in fact we discussed the potential metabolic pathways of the metabolites found altered in BCa within each sample type, and considering the information described in the original papers, integrating the compounds from the same metabolic processes as well as the implications of their alterations in cancer progression. Then, our aim was to integrate those metabolites found consistently modulated by BCa independently of sample type into metabolic pathways. To achieve this, we used the bioinformatic tool MetScape, which relies on the KEGG IDs of the metabolites. Consequently, this analysis was limited to the 6 out of 9 metabolites for which KEGG IDs were available. The analysis revealed distinct molecular pathways associated with these metabolites. As there were no connections between these pathways, we opted not to further expand the discussion at the pathway level. We believe that the significance of the pathways involving these metabolites modulated by BCa warrants further exploration in future studies. Nevertheless, in line with the suggestions from Reviewer #1, we have further clarified the analysis conducted with MetScape in Section 3, and we have included the pathways in which the metabolites participate in the revised version of Supplementary Figure S2. Additionally, we have made some amendments in Section 4 to enhance the clarity of the proposed metabolite signature for BCa.
3) Statistical methods are not described in detail for the mentioned studies. Are they the same or comparable? What about preprocessing the metabolic data in the studies mentioned, e. g., was any uniform normalization there?
R: The selection of the metabolites listed in Supplementary Table 2 was based on the information extracted from the original papers, with no additional statistical analysis conducted. In the context of our review, we evaluated the metabolite levels in terms of increase or decrease (+ or – as presented in column “Trend” of Supplementary Table 2) without considering absolute values, when available. To avoid misinterpretation, the revised version of the manuscript includes the following sentence in section 3: “The variation trends of the metabolites outlined in this table align with those reported in the original papers.”
Round 2
Reviewer 2 Report
Comments and Suggestions for Authors
The authors changed type of the article, now the type is appropriate. Two other points have been improved insufficiently.
2) I thank the authors for metascape analysis. Hovewer, information about the pathways described must be included in the manuscript.
To avoid misleading, pathways sources or structures are needed, because usually there are several pathways correspond to one name in different sources. Then, number and percent of measured metabolites must be described, like here https://bmccancer.biomedcentral.com/articles/10.1186/s12885-022-09318-5/tables/3
It is very important to highlight that significantly changed matabolites are included in pathways, but we cannot assess total pathways changes. Moreover, many studies claimed that pathways are significantly changed without providing of p value or despite of p-values(e.g.,ref 17)
It is misleading information that has to be corrected in review.
3. Please, provide supplementary table with statistical methods for each described study
Author Response
The authors gratefully acknowledge the constructive comments from Reviewer #2 and the meticulous revisions made to our paper, which significantly enhanced its scientific quality.
The authors changed type of the article, now the type is appropriate. Two other points have been improved insufficiently.
2) I thank the authors for metascape analysis. However, information about the pathways described must be included in the manuscript.
R: Based on the reviewer's suggestion, we have incorporated Figure S1 into the manuscript, now denoted as Figure 6. Additionally, the pathways have been elucidated in the revised version of the manuscript. As a result, the manuscript now provides comprehensive insights into: “Furthermore, only one metabolite from each pathway was consistently found in urine and BCa tissue or cells (Figure 6). Vitamin B5-CoA biosynthesis from pantothenate and porphyrin metabolism are found both downregulated, considering the metabolites involved whereas arachidonic acid metabolism, pyrimidine metabolism, BCAA metabolism, and fatty acid metabolism are upregulated. Nevertheless, CoA, derived from pantothenate, assumes a central role in bridging these pathways by acting as a cofactor for crucial enzymatic reactions involved in fatty acid metabolism, amino acid metabolism, and porphyrin metabolism. However, the downregulation of CoA biosynthesis seems to correlate with the downregulation of porphyrin metabolism but with the upregulation of fatty acid metabolism, particularly arachidonic acid, and BCAA metabolism, suggesting the involvement of complex regulatory signaling pathways.”
To avoid misleading, pathways sources or structures are needed, because usually there are several pathways correspond to one name in different sources. Then, number and percent of measured metabolites must be described, like here https://bmccancer.biomedcentral.com/articles/10.1186/s12885-022-09318-5/tables/3
It is very important to highlight that significantly changed matabolites are included in pathways, but we cannot assess total pathways changes. Moreover, many studies claimed that pathways are significantly changed without providing of p value or despite of p-values (e.g., ref 17).
It is misleading information that has to be corrected in review.
R: In response to the Reviewer's suggestion, which we acknowledge, we have introduced a new supplementary table (now Table S8) to complement the information presented in Figure 6. This new table provides detailed insights into the metabolites associated with each pathway, along with their corresponding p-values and false discovery rates, obtained from MetScape analysis. Notably, one metabolite for each metabolic pathway was sourced from our integrated literature analysis. Hence, we have chosen to include this supplementary information in Table S8 rather than within the main manuscript. Additionally, in the legend of Figure 6, we have included the following sentence: “The comprehensive details of pathway enrichment, including the associated metabolites for each pathway along with their corresponding FDR and p-values, are presented in Table S8.”
- Please, provide supplementary table with statistical methods for each described study.
R: In response to the reviewer's recommendation, we have incorporated a new column titled "Statistical Methods" into Supplementary Table S1, elucidating the methodologies utilized in each study for the analysis of metabolic data.